# Lactobacillus Reuteri DSM 17938 *(Limosilactobacillus reuteri)* in Diarrhea and Constipation: Two Sides of the Same Coin?

**DOI:** 10.3390/medicina57070643

**Published:** 2021-06-23

**Authors:** Angela Saviano, Mattia Brigida, Alessio Migneco, Gayani Gunawardena, Christian Zanza, Marcello Candelli, Francesco Franceschi, Veronica Ojetti

**Affiliations:** 1Department of Emergency Medicine, Università Cattolica del Sacro Cuore, 00168 Rome, Italy; gayani-g@hotmail.com; 2Department of Gastroenterology, Università Tor Vergata, 00133 Rome, Italy; mattiabrigida@hotmail.it; 3Department of Emergency Medicine, Fondazione Policlinico Universitario A. Gemelli, IRCCS, 00168 Rome, Italy; alessio.migneco@policlinicogemelli.it (A.M.); marcello.candelli@policlinicogemelli.it (M.C.); francesco.franceschi@policlinicogemelli.it (F.F.); veronica.ojetti@policlinicogemelli.it (V.O.); 4Department of Anesthesiology, Critical Care and Emergency Medicine-Fondazione Nuovo Ospedale Alba-Bra, Michele and Pietro Ferrero Hospital, 12060 Verduno, Italy; christian.zanza@live.it

**Keywords:** *Lactobacillus reuteri* (DSM 17938), diarrhea, constipation, *Limosilactobacillus reuteri*, emergency department, probiotics

## Abstract

*Background and Objectives*: *Lactobacillus reuteri* DSM 17938 (*L. reuteri*) is a probiotic that can colonize different human body sites, including primarily the gastrointestinal tract, but also the urinary tract, the skin, and breast milk. Literature data showed that the administration of *L. reuteri* can be beneficial to human health. The aim of this review was to summarize current knowledge on the role of *L. reuteri* in the management of gastrointestinal symptoms, abdominal pain, diarrhea and constipation, both in adults and children, which are frequent reasons for admission to the emergency department (ED), in order to promote the best selection of probiotic type in the treatment of these uncomfortable and common symptoms. *Materials and Methods*: We searched articles on PubMed^®^ from January 2011 to January 2021. *Results*: Numerous clinical studies suggested that *L. reuteri* may be helpful in modulating gut microbiota, eliminating infections, and attenuating the gastrointestinal symptoms of enteric colitis, antibiotic-associated diarrhea (also related to the treatment of Helicobacter pylori (HP) infection), irritable bowel syndrome, inflammatory bowel disease, and chronic constipation. In both children and in adults, *L. reuteri* shortens the duration of acute infectious diarrhea and improves abdominal pain in patients with colitis or inflammatory bowel disease. It can ameliorate dyspepsia and symptoms of gastritis in patients with HP infection. Moreover, it improves gut motility and chronic constipation. *Conclusion*: Currently, probiotics are widely used to prevent and treat numerous gastrointestinal disorders. In our opinion, *L. reuteri* meets all the requirements to be considered a safe, well-tolerated, and efficacious probiotic that is able to contribute to the beneficial effects on gut-human health, preventing and treating many gastrointestinal symptoms, and speeding up the recovery and discharge of patients accessing the emergency department.

## 1. Introduction

*Lactobacillus reuteri* DSM 17938, recently renamed *Limosilactobacillus reuteri* (*L. reuteri*), is a probiotic well-identified for its beneficial effects on some gastrointestinal diseases [1,2,3]. The mother strain, ATCC 55730, a clinically proven probiotic, was isolated around 1990 [4]. This strain was later cured from two plasmids, which generated the strain that is used today, *L. reuteri* DSM 17938, obtained through the removal of antibiotic resistance gene-carrying plasmids from *Lactobacillus reuteri* ATCC 55730 [5]. It belongs to the genus *Lactobacillus*, which includes many other gram-positive oxygen tolerant fermentative bacteria such as *L. acidophilus*, *L. bulgaricus*, *L. casei* and *L. rhamnosus* [6].

*L. reuteri* can colonize different human body sites, including primarily the gastrointestinal tract, but also the vagina [1]. Recently, the interest in this probiotic has significantly increased thanks to its properties in the prevention and in the treatment of numerous gastrointestinal symptoms and disorders, both in children and in adults. *L. reuteri* adheres to the intestinal epithelium, producing proteins able to bind with the mucus, making it tough for pathogen microorganisms [7] to enter, and thereby remodeling the balanced composition of gut microbiota, as demonstrated by a study on pigs conducted by Hou et al. [7]. Moreover, *L. Reuteri* produces antimicrobial molecules and promotes the development and the functionality of regulatory T cells, strengthening the gut barrier, and decreasing the microbial translocation from the intestinal lumen to the tissue, as reported by studies conducted on animal models [7], on humans [8] and in vitro [9]. Literature preclinical studies [10], using genetic tools such as genome sequencing, molecular tools, and genomic-based approaches (both in mice and then in humans) showed that *L. reuteri* has multiple beneficial effects on gastrointestinal symptoms, on gut infections, HP eradication, antibiotic-associated diarrhea, inflammatory bowel syndrome (IBS), inflammatory bowel disease (IBD), and colorectal cancer [11,12,13]. It can reduce abdominal pain in infantile colic, the functional abdominal discomfort in children, and it can decrease crying due to necrotizing enterocolitis in preterm neonates [12,13,14]. It can improve gut motility and chronic constipation as demonstrated in infants [2], and in vitro and mice studies [15], with beneficial effects on patients’ disorders. The aim of this review is to summarize the current knowledge on the properties of *L. reuteri* in the management of gastrointestinal disorders, namely diarrhea and constipation, which are frequent reasons for admission to the ED, in order to promote the best selection of probiotic type, directly in the emergency setting, for the treatment of these uncomfortable and common symptoms.

### 1.1. L. reuteri for Treating Acute Watery Diarrhea 

Numerous clinical studies have been conducted to explore the function of *L. reuteri* in the intestines of healthy individuals, its role in regulating gut microbiota and mucosal homeostasis, in shaping the intestinal host immune system, and in ameliorating intestinal inflammation in pathological conditions, such as acute watery diarrhea [8,9,10,16,17,18,19] (Appendix A). Literature data provides evidence that the use of lactobacilli leads to an improvement of gut functionality and gastrointestinal symptoms, as reported by Guandalini et al. [20]. *L. reuteri* restores the balanced composition of human microbiota communities, and is useful both in the treatment of acute watery diarrhea and in the prevention of new episodes of diarrhea, including after prolonged antibiotic treatments [21,22]. Shornikova et al. [23,24] examined the role of *L. reuteri* in acute watery diarrhea in children and in rotavirus gastroenteritis. These authors conducted a randomized controlled clinical trial, enrolling 86 children, between 6 and 36 months of age, who tested positive for rotavirus. They randomized children to receive either 10^10^ or 10^7^ colony-forming units (CFU) of *L. reuteri* or a placebo once a day for 5 days. They showed that the use of *L. reuteri* shortened the duration of the acute watery diarrhea with a dose-related effect. Indeed, the mean duration of acute watery diarrhea was 1.5 days in the group taking a large dosage of *L. reuteri*, 1.9 days in the group taking a small dosage, and 2.5 days in the group taking the placebo. By the second day of treatment with *L. reuteri*, the acute watery diarrhea persisted among 48% of those who took the large dosage, 70% of those who took the small dosage, and 80% of those treated with the placebo. Francavilla et al. [18], in their randomized placebo-controlled clinical trial, which included 35 children in the *L. reuteri* group and 34 in the placebo group, reported that supplementation with *L. reuteri*, at a dosage of 4 × 10^8^ CFU/day for 7 days, reduced the duration of acute watery diarrhea, with the maximum effect on the second and third day, in children aged between 3 months and 3 years, without reported side effects. Dinleyici et al. [19,25] carried out two multicenter randomized clinical trials and found that the use of *L. reuteri* was able to decrease the duration of acute watery diarrhea up to 15 h in children aged between 3 months and 5 years. Moreover, it was able to reduce the length of hospital stay. After two days of treatment with *L. reuteri*, ~55% of children were diarrhea-free vs. only 15% of children in the control group, with a greater effectiveness of the intervention between 48 and 72 h after using five drops containing 10^8^ CFU *L. reuteri*, and with a safe, well-tolerated and effective profile in the pediatric outpatient setting [19]. Urbańska et al. [26], in their review, highlighted that the frequency of diarrhea in children (they reviewed published articles and trials for a total of 1229 participants) treated with *L. reuteri* at a dosage ranging from 1 × 10^8^ to 4 × 10^8^ CFU daily for 5–7 days, was surprisingly low. *L. reuteri* was able to reduce the duration of diarrhea by one day with the maximum beneficial effect at day two. Even if the studies they analyzed were heterogeneous for the duration and the dosage of *L. reuteri,* the authors confirmed the beneficial effects of this probiotic for the treatment and prevention of acute watery diarrhea. Szymański et al. [27] concluded that *L. reuteri*, at a dosage of 2 × 10^8^ CFU for 5 days, could help in the management of acute watery diarrhea in children, shortening the duration of hospitalization. They collected data on about 99 children aged <5 years with acute gastroenteritis (liquid stool) lasting no longer than five days, and an increase in the frequency of evacuations (≥3 evacuations/day), demonstrating that the administration of *L. reuteri* vs. a placebo, in addition to standard rehydration therapy, reduced the duration of hospitalization, but not the duration of diarrhea, which was similar in both groups. Margiotta et al. [28] showed that in children with acute gastroenteritis, the combination of *Lactobacillus reuteri* LRE02-DSM 23878, at a dose of 2 × 10^8^ CFU/daily, and *Lactobacillus rhamnosus* LR04-DSM 16605, at a dose of 1 × 10^9^ CFU/daily for 15 days, improved stool consistency and the number of evacuations. Patro-Golab et al. [29] performed a systematic review and meta-analysis analyzing four relevant trials that compared the administration of *L. reuteri*, at different dosages with a placebo or without treatment. Their primary endpoints were the diarrhea duration and the stool volume. Their secondary endpoint was the evaluation of the effects of *L. reuteri* on the course of diarrhea, on the duration of diarrhea no longer than seven days, and on the duration of hospitalization. They observed that *L. reuteri* was useful in reducing diarrhea duration by approximately 21 h and hospitalization in children by approximately 13 h, but no significant effects were reported on the number of watery stools. The authors concluded that probiotics and *L. reuteri* could be a useful and safe, supportive measure for the treatment and prevention of diarrhea, reducing both the diarrhea duration and the intensity of symptoms, with beneficial health effects [29]. 

### 1.2. L. reuteri and Gastrointestinal Symptoms Related to Antibiotics HP Eradication-Treatment

*L. reuteri* has many anti-inflammatory proprieties [9]. It produces reuterin, which is a potent anti-microbic compound able to inhibit the growth of gram-positive and gram-negative bacteria, fungi, and protozoa [30]. Moreover, *L. reuteri* forms a biofilm rich in probiotic functions, inhibits the production of proinflammatory cytokines, and prevents intestinal overgrowth by other commensals, thereby maintaining a balanced gut-environment [30]. Furthermore, it contributes to the restoration of the balanced composition of gut microbiota after an antibiotic treatment. For example, antibiotic treatment is essential to treat HP infection, but it can lead to dysbiosis, alteration of gut microbiota composition, and reduction of the richness and wealth of microflora strains, with subsequent diarrhea and gastrointestinal discomfort. *L. reuteri* can improve gastrointestinal symptoms and reduce antibiotic side effects, restoring the balance of gut microflora and allowing patients to be more compliant with their antibiotic therapy. Ojetti et al. [31] determined that *L. reuteri* was effective in HP eradication and, in particular, in the prevention of gastrointestinal symptoms (abdominal pain, diarrhea, nausea, vomiting, and bloating) associated with the use of antibiotics for the treatment of HP infection. Dore et al. [12] proved that *L. reuteri* could improve gastrointestinal symptoms related to HP infection. It prevented the HP colonization of the human gut mucosa by inhibiting HP-binding to the glycolipid receptors [12]. Moreover, it increased the production of mucin, reuterin, and antioxidant substances, stabilized the mucosal barrier, and stimulated the mucosal immunity [12,32] with beneficial health effects on dysbiosis of gut microbiota after the use of antibiotics and antisecretory treatments (the standard of care to eradicate HP). Wang et al. [33] recommended the use of *Lactobacillus* species to treat HP infection and gastrointestinal symptoms (dyspepsia, diarrhea, and abdominal pain) related to antibiotics administration, as recommended for the treatment of HP infection by current guidelines [13]. Jones et al. [9] assessed that *L. reuteri* promoted the biofilm formation and the colonization of gut mucosa by commensal lactobacilli, establishing a protective niche and preventing both the invasion of opportunistic bacteria and the gastrointestinal side effects related to the use of antibiotics in the treatment of some gastrointestinal diseases. Emara et al. [34,35] concluded that *L. reuteri*, through different mechanisms, including the production of reuterin, showed an effective action against HP and improved its eradication rate, the clinical and pathological features, and symptoms related to this infection. Buckley et al. [36] demonstrated that *L. reuteri* suppressed HP infection, improving gastrointestinal symptoms in patients affected by this infection, and ameliorating antibiotic-gastrointestinal side effects that are common after HP eradication-treatment with antibiotics. 

### 1.3. L. reuteri for Treating Constipation in Children

Chronic constipation is a common gastrointestinal disorder that can affect patients of all ages and severely impact their quality of life [37]. Literature studies showed that in chronic constipation there is an alteration of the gut microbiota that could be restored with some probiotics strains as *L. reuteri* (Appendix A). The latter showed the ability to produce short chain fatty acids (SCFA), to reduce the gut intraluminal level of pH, and to promote the colonic peristalsis, influencing the frequency and velocity of colonic myoelectric cells with beneficial effects on chronic constipation [37]. Current evidence reported that *L. reuteri* improved bowel movements in patients (both children and adults) with chronic constipation [2,3], but did not affect stool consistency [37]. Kubota et al. [2] reported that *L. reuteri* administered to children with chronic constipation, twice a day for four weeks, induced changes in the composition of the gut microbiota (reducing Clostridiales genera, such as Oscillospira, Megasphaera, and Ruminococcus), enhancing intestinal motility and reducing the transit time of stool, with significant results in the fourth week. *L. reuteri* improved the stool frequency, but not the stool consistency [2]. Coccorullo et al. [38] proved that *L. reuteri* had a positive effect on functional constipation in infants, ameliorating the bowel frequency at week 2, week 4, and week 8 of administration. Constipation in infants is often related to the changes in diet (for example, the passage from breast milk to commercial formula or the introduction of solid food) with important changes in gut microbiota composition (reduction of Bifidobacteria and Lactobacilli) that *L. reuteri* helps to restore. Indrio et al. [39] underlined that *L. reuteri* reduced constipation during the first 3 months of life. Early life events could alter the balance of gut microbiota, increasing visceral sensitivity and mucosal permeability, which can be restored via the administration of Lactobacilli. Jadresin et al. [40] added that there was no benefit derived from the treatment with *L. reuteri* in children with constipation. Gomes et al. [41] studied fecal microbiota composition during constipation, but they did not find a specific pattern and argued that, although probiotics could have positive effects on intestinal functionality, their use was not yet recommended by literature data in the treatment of constipation in children. Indeed, the composition of the intestinal microbiota in children with constipation before and after the probiotic administration has not been evaluated yet. According to these authors, more studies are needed to investigate the use of probiotics in constipation and the mechanisms by which *L. reuteri* modulates gut motility with effects on constipation in children [15,41].

### 1.4. L. reuteri for Treating Constipation in Adults

The use of probiotics in organic and functional adult gastrointestinal disorders has gained a growing interest. It is known that the gut microbiota influences the intestinal motility through the fermentation of both carbohydrates and protein, and through the production of SCFA, such as butyrate, acetate, and propionate, and gases, such as H_2_ and CO_2_, which can influence the gut smooth muscle motility and microbiota composition [37]. Ojetti et al. [3] highlighted that the administration of *L. reuteri* twice a day for four weeks was effective in reducing methane (CH_4_) production by gut microflora (Methanobrevibacter smithii), with an increase in bowel movements and an improvement in chronic constipation. Riezzo et al. [42] showed the beneficial effect of *L. reuteri* for defecation and for symptoms of abdominal discomfort, pain, and bloating due to the modulation of the serum levels of serotonin (5-HT) and brain-derived neurotrophic factor (BDNF) by this probiotic strain. The 5-HT pathways play a pivotal role in the interaction between gut microbiota and the enteric nervous systems, with health benefits (which are not completely fully understood) on gut motility and constipation. The authors included 56 patients with constipation, randomized to receive *L. reuteri* for 105 days. Dimidi et al. [43,44], in their review on the mechanisms of action of probiotics on the gut motility and constipation, and the interaction with the gastrointestinal microbiota, demonstrated that *L. reuteri* interacts with the gut-brain axis and modulates the afferent sensory nerves that influence gut motility. Moreover, it involves the enteric nervous system, increasing the excitability of myenteric neurons in rats through action on the 5-HT pathways. The latter is produced by the enteric nervous system and it is a key neurotransmitter with an essential role for the stimulation of the local enteric nervous reflexes and the promotion of gut secretion and propulsive motility, improving constipation and gastrointestinal related disorders. West et al. [15] showed that the administration of *L. reuteri* to adult mice promoted the reduction of jejunal motility, but increased that in the colon, with beneficial effects on constipation. On the contrary, Wegh et al. [45] argued that currently available evidence was insufficient to promote the use of probiotics in functional constipation. Therefore, more evidence is needed in order to fully understand the action of *L. Reuteri* on intestinal motility, abdominal discomfort, pain, and bloating, and, consequently, on functional constipation, and to recommend it as a “standardized” treatment. 

## 2. Materials and Methods 

We searched articles on PubMed^®^ from January 2011 to January 2021. No ethical approval was necessary to perform this review. The principal words we included were *Lactobacillus reuteri* DSM 17938 (*L. reuteri*), *L. reuteri* ATCC 55730, *Limosilactobacillus reuteri*, diarrhea, antibiotics related-diarrhea, gastroenteritis, probiotics, colitis, gut microbiota, gastrointestinal symptoms, constipation, gut motility, and Helicobacter pylori (HP). We combined these issues in our research, excluding other strains of probiotics, or other strains of *Lactobacillus reuteri*. We ruled out the use of this probiotic in other human disorders as well. We focused on the reliable and specific use of *L. reuteri* in the main gastrointestinal disorders we meet in the ED, namely diarrhea and constipation. 

## 3. Results 

This review suggests that not all types of probiotics have an equal effect on the management of gastrointestinal diseases. In particular, we collected data on *L. reuteri* which revealed it to be a helpful probiotic in modulating gut microbiota; it promotes the development and the functionality of regulatory T cells, produces reuterin that inhibits the growth of some harmful gram-negative and gram-positive bacteria, along with yeasts, fungi, and protozoa [24]. Moreover, it survives bile and gastric acid and colonizes the gut microbiota [1]. We focused our attention on its role in acute diarrhea, both in adults and children. Many authors concluded that *L. reuteri* supplementation of more than 10^7^ CFU was able to shorten the duration of acute watery diarrhea with a dose-related effect, with the maximum effect on the second and third day, thus reducing the length of hospital stay. No side effects were reported in any studies. It is well known that antibiotics can lead to dysbiosis, alteration of the gut microbiota composition, reduction of the richness and wealth of microflora strains, with subsequent diarrhea and gastrointestinal discomfort. Many studies suggested that supplementation with *L. reuteri* during antibiotic therapy, for example against HP, was useful in reducing gastrointestinal side effects with an increase in the eradication rate. This specific probiotic prevents the HP colonization of the human gut mucosa and inhibits HP-binding to the glycolipid receptors [12]. Moreover, it increased the production of mucin and of antioxidant substances, stabilized the mucosal barrier, and stimulated the mucosal immunity. Reuterin directly inhibits HP growth. On the other hand, we analyzed data on the role of *L. reuteri* in chronic constipation, both in adult and children. Literature supported the evidence that *L. reuteri* supplementation, even if for a few weeks, improved bowel movements and stool consistency. The possible explanation could be that *L. reuteri* increased both frequency of colonic myoelectric motility complex and velocity. Another possible explanation could be an enhanced water and electrolyte secretion due to probiotic activity. Some authors [3] hypothesized that *L. reuteri* may exert a beneficial effect by reducing the gut methanogenic flora and increasing bowel movements. A high level of gut microflora with a CH4 production is associated with a delayed orocecal transit time and, consequently, constipation. The ability of *L. reuteri* to inhibit gut microflora with an overexpression of H2 consumption, and a consequent reduction of CH4 methane production, could represent the base for new and more effective therapies.

## 4. Discussion

The use of probiotics in human health and diseases has increased in recent years [46]. Literature data shows conflicting results. Many authors agree that not all types of probiotics are equal in the management of gastrointestinal disorders [47,48], not all type of probiotics products have been validated [48,49], and there is a deep variability in commercially available probiotics formulation [50]. Gastrointestinal disorders are a common cause of admission to medical services, such as the ED. In particular, diarrhea and constipation are frequent reasons for medical consultations. In our opinion, they are two sides of the same coin. In fact, both diarrhea and constipation are the consequence of dysbiosis, alteration in human gut microbiota composition, and dysregulation of the complex interaction between microbe communities and host immune systems [51]. Probiotics are live microorganisms used as nutritional supplements [50,51,52] that can restore the intestinal health and ameliorate the gastrointestinal symptoms (such as abdominal pain, diarrhea, and constipation). Emerging evidence showed that probiotic activities and effects are strain and dose related [53,54]. *L. reuteri* has been identified as part of the normal human gut population and its administration as a “probiotic” has been recommended in gut disorders related to infections with effective results [2,16,55]. *L reuteri* is the most widely studied probiotic in children, with concentrations ranging from 10^6^ CFU/daily up to 10^8^ to 10^9^ CFU/daily [16,17,56]. *L. reuteri* has various mechanisms of action that are not fully understood. It stimulates the mucosal gut barrier function well, produces antimicrobial substances (such as reuterin and lactic acid), and influences the local acquired and innate immune response [46,47,57,58]. It improves antibiotic related-diarrhea and infectious diarrhea, inhibiting the growth of rotavirus in infants, and the growth of *E. coli*, Salmonella, Clostridium difficile, and Campylobacter jejuni in vitro and in vivo studies [23,30,58,59,60]. Moreover, *L. reuteri* survives gastric and bile juice and restores intestinal epithelial cell morphology after damages caused by pathogens [15,16]. It ameliorates digestive disorders, gastritis, dyspepsia, nausea or vomiting by acting against gut pathogens [31,32]. It improves gut motility, ameliorates constipation, and bloating by reducing methane (CH4) production via the Methanobrevibacter smithii species, increases the excitability of myenteric neurons (in rat models), and modulates the action of the serum levels of 5-HT and BDNF in the gut. It has a safe and healthy profile and can be promptly used for its prophylactic and therapeutic effects on the human gut [49]. 

## 5. Conclusions

In conclusion, *L. reuteri* plays a key role in maintaining a balanced microbiota gut composition. Many clinical trials have proved the safety, efficacy and tolerance of this probiotic in preventing and treating numerous gastrointestinal disorders, ranging from diarrhea to constipation. In our opinion, *L. reuteri* meets all the requirements to contribute to beneficial effects on human-gut health in the emergency setting. In fact, gastrointestinal disorders, as stated previously, are one of the most common causes of admission to the ED. The use of *L. reuteri*, directly in the emergency room can speed up the remission of symptoms, reduce the recovery period, and promote the safe discharge of patients.

## Data Availability

Data sharing not applicable.

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
