# Peer review of "Lactobacillus Reuteri DSM 17938 (Limosilactobacillus reuteri) in Diarrhea and Constipation: Two Sides of the Same Coin?"

_medicina, 2021, doi:10.3390/medicina57070643_

Round 1

Reviewer 1 Report

This review with the aim to summarize the current knowledge of the role of L. reuteri in the management of gastrointestinal symptoms both in adults and children is overall well written, concise and complete, therefore it would benefit only of some minor revisions:

  1. The introduction, is interesting and adequately written
  2. For “1. L. Reuteri for treating acute watery diarrhea”, in the pediatric population part, we point out the presence of a recent survey in favor of your statements, which is “Effect of Lactobacillus reuteri LRE02-Lactobacillus rhamnosus LR04 combination and gastrointestinal functional disorders in an Emergency Department pediatric population” by Margiotta G. et al, released in April 2021.
  3. For “1.4. L. Reuteri for treating constipation in adults”, maybe a concluding sentence is required to better summarize the different statements about the use of L.Reuteri in this population.
  4. For “Materials and methos”, no other inclusion criteria nor exclusion criteria were applied?
  5. For ”Results”, could you please be more precise and discursive about the results highlighted?
  6. Discussion and Conclusions are well written.

In conclusion, this article is interesting, useful and well written, but Authors have to clarify the above points.

Author Response

  1. For “1. L. Reuteri for treating acute watery diarrhea”, in the

pediatric population part, we point out the presence of a

recent survey in favor of your statements, which is “Effect of

Lactobacillus reuteri LRE02-Lactobacillus rhamnosus LR04

combination and gastrointestinal functional disorders in an

Emergency Department pediatric population” by Margiotta G.

et al, released in April 2021.

We added this interesting paper as suggested

  1. For “1.4. L. Reuteri for treating constipation in adults”, maybe

a concluding sentence is required to better summarize the

different statements about the use of L. Reuteri in this

population.

 We added a conclusion statement as suggested

  1. For “Materials and methods”, no other inclusion criteria nor

exclusion criteria were applied?

We searched paper combining these issues; we excluded different strains of Lactobacilli or the use of L. reuteri in other human disorders, focusing on diarrhea and constipation (we completed in the manuscript).

  1. For “Results”, could you please be more precise and

discursive about the results highlighted?

We modified results as suggested

Reviewer 2 Report

This review paper summarizes some of the work that have been conducted on the nowadays well studied probiotic strain L. reuteri DSM 17938. Well written reviews are valuable and there is a need to enlighten different aspects of this fascinating species. So it could potentially be an important review, but it doesn’t feel completely elaborated and contains a little too many errors. I think it needs extensive modifications before it could be accepted for publication.

Major comments

- I think the structure should be changed. The result section is now around five lines and doesn’t say much. But the result from the literature review is mainly found in the Introduction and is difficult to distinguish from the real introduction.

- In several cases it’s difficult to link the references to the described results and sometimes results from in vitro studies are mixed with animal studies and human interventions. I think this should be worked through properly.

Comments on details

- Abstract and line 45-46. To my knowledge L. reuteri has never been isolated from skin. Urinary tract is not really correct, but you should write vagina. I lack references for this section.

- Line 41-42. L. reuteri DSM 17938 was not isolated 1962. It’s mother strain ATCC 55730 was isolated around 1990 (Connolly, E. (2004). Lactobacillus reuteri ATCC 55730: a clinically proven probiotic. NutraFoods 3(1) 15-22. But this strain was later cured from two plasmids which generated the strain that is used today (DSM 17938; Rosander, A., Connolly, E., & Roos, S. (2008). Removal of antibiotic resistance gene-carrying plasmids from Lactobacillus reuteri ATCC 55730 and characterization of the resulting daughter strain, L. reuteri DSM 17938. Applied and Environmental Microbiology, 74(19), 6032–6040. http://doi.org/10.1128/AEM.00991-08)

- Line 44. Lactobacillus are not really anaerobes. They could be described to be oxygen tolerant fermentative bacteria.

- The name Limosilactobacillus reuteri is just mentioned in one sentence. This is the correct name and I think it should be used in the title and other places. Better to mention that it previously was named Lactobacillus reuteri. Also, it would be good with a reference.

- Abstract. Background and Objectives describes many effects that should be part of Results.

- All superscript should be looked through. E.g. 1010 cfu should be 10^10 cfu (several places)

- It would be good to summarize the studies that were found in the literature review in a table.

- Line 250. What is meant with “minimum concentration of 10^6 CFU/ml or gram”?

Author Response

- I think the structure should be changed. The result section is

now around five lines and doesn’t say much. But the result from

the literature review is mainly found in the Introduction and is

difficult to distinguish from the real introduction.

We modified result section as suggested

- In several cases it’s difficult to link the references to the

described results and sometimes results from in vitro studies are

mixed with animal studies and human interventions. I think this

should be worked through properly.

We described results differentiating in vitro studies, animal studies, human interventions.

Comments on details

- Abstract and line 45-46. To my knowledge L. reuteri has never

been isolated from skin. Urinary tract is not really correct, but

you should write vagina. I lack references for this section.

We corrected as suggested, introducing the reference

- Line 41-42. L. reuteri DSM 17938 was not isolated 1962. It’s mother strain ATCC 55730 was isolated around 1990 (Connolly, E. (2004). Lactobacillus reuteri ATCC 55730: a clinically proven probiotic. NutraFoods 3(1) 15-22. But this strain was later cured from two plasmids which generated the strain that is used today (DSM 17938; Rosander, A., Connolly, E., & Roos, S. (2008). Removal of antibiotic resistance gene-carrying plasmids from Lactobacillus reuteri ATCC 55730 and characterization of the resulting daughter strain, L. reuteri DSM 17938. Applied and Environmental Microbiology, 74(19), 6032–6040.

We corrected and added these studies suggested

- Line 44. Lactobacillus are not really anaerobes. They could be

described to be oxygen tolerant fermentative bacteria.

We modified as suggested

- The name Limosilactobacillus reuteri is just mentioned in one

sentence. This is the correct name and I think it should be used

in the title and other places. Better to mention that it previously

was named Lactobacillus reuteri. Also, it would be good with a

reference.

We added it in the title as suggested, in the manuscript we used abbreviation L. reuteri

- Abstract. Background and Objectives describes many effects

that should be part of Results.

We modified as suggested and we included them in the results 

- All superscript should be looked through. E.g. 1010 cfu should

be 10^10 cfu (several places)

We corrected in the manuscript as suggested

- It would be good to summarize the studies that were found in the literature review in a table.

As suggested, we summarized the main studies in a table

- Line 250. What is meant with “minimum concentration of 10^6

CFU/ml or gram”?

We clarified this sentence  in the manuscript